# Lignosulphonates as an Alternative to Non-Renewable Binders in Wood-Based Materials

**DOI:** 10.3390/polym13234196

**Published:** 2021-11-30

**Authors:** Sofia Gonçalves, João Ferra, Nádia Paiva, Jorge Martins, Luísa H. Carvalho, Fernão D. Magalhães

**Affiliations:** 1LEPABE–Faculdade de Engenharia da Universidade do Porto, Rua Dr. Roberto Frias, 4200-465 Porto, Portugal; up201808942@edu.fe.up.pt (S.G.); jmmartins@estgv.ipv.pt (J.M.); lhcarvalho@estgv.ipv.pt (L.H.C.); 2Sonae Arauco Portugal S.A., Lugar do Espido—Via Norte, 4470-177 Porto, Portugal; joao.ferra@sonaearauco.com (J.F.); nadia.paiva@sonaearauco.com (N.P.); 3DEMad–Departamento de Engenharia de Madeiras, Instituto Politécnico de Viseu, Campus Politécnico de Repeses, 3504-510 Viseu, Portugal

**Keywords:** lignosulphonates, lignin, formaldehyde, wood adhesives

## Abstract

Lignin is a widely abundant renewable source of phenolic compounds. Despite the growing interest on using it as a substitute for its petroleum-based counterparts, only 1 to 2% of the global lignin production is used for obtaining value-added products. Lignosulphonates (LS), derived from the sulphite pulping process, account for 90% of the total market of commercial lignin. The most successful industrial attempts to use lignin for wood adhesives are based on using this polymer as a partial substitute in phenol-formaldehyde or urea-formaldehyde resins. Alternatively, formaldehyde-free adhesives with lignin and lignosulphonates have also been developed with promising results. However, the low number of reactive sites available in lignin’s aromatic ring and high polydispersity have hindered its application in resin synthesis. Currently, finding suitable crosslinkers for LS and decreasing the long pressing time associated with lignin adhesives remains a challenge. Thus, several methods have been proposed to improve the reactivity of lignin molecules. In this paper, techniques to extract, characterize, as well as improve the reactivity of LS are addressed. The most recent advances in the application of LS in wood adhesives, with and without combination with formaldehyde, are also reviewed.

## 1. Introduction

Lignin is a complex, amorphous, natural polymer, and one of the most abundant in nature, only behind cellulose [1]. Although lignin is the main by-product of the paper pulping processes, it is usually burned as fuel [2].

Out of the 50 to 70 million tons of the lignin that is produced annually, only 1 to 2% is actually used for the production of value-added products. Therefore, it can be concluded that lignin is an underutilized material [3].

In plants, cellulose is present in the form of bundled fibrils that are bound together by hemicellulose and lignin, providing rigidity to the cell wall [4]. Lignin performs the following functions in plants: providing mechanical support, slowing down decomposition, forming a barrier towards water evaporation, and helping to transport water to vital areas of the plant [2]. In vitro studies have shown that lignin and lignin extracts display antimicrobial and antifungal activity, act as antioxidants, absorb UV radiation, and exhibit flame-retardant properties [5]. 

Lignin is the main renewable source of phenolic compounds of natural origin. This has been increasingly researched as a more eco-friendly alternative to the petroleum-based counterparts [6]. The molecule is comprised of several types of methoxylated phenylpropanoid units (C9). The three primary precursors of lignin are: *p*-coumaryl, coniferyl, and sinapyl alcohols, as shown in Figure 1. These monolignols are also known as *p*-hidroxyphenyl (H), guaiacyl (G), and syringyl (S) units, respectively [6,7,8].

These monolignols differ in the number of methoxy groups that are attached to the aromatic moiety: sinapyl alcohol has two methoxy groups, coniferyl alcohol has one methoxy group, and *p*-coumaryl alcohol has none [6]. Different types of plants display different degrees of participation of the major monolignols. As shown in Table 1, the main monolignols in softwood and hardwoods are coniferyl and sinapyl alcohol, respectively [9].

Researchers admit that no plants contain lignin that is only derived from the three primary precursors [6]. Noncanonical subunits that have been identified include caffeoyl alcohol, which was discovered in the seeds of some Vanilla and Cactaceae species [10]. Other subunits are ferulic acid, ferulates (which form linkages between hemicellulose and lignin), coniferaldehyde, sinapaldehyde, and acylated monoglinols [6]. Intermediate free radicals are generated from these lignin precursors through the dehydrogenation of phenolic OH groups by the plant’s peroxidase and laccase enzymes. The polymerization process then occurs as follows: firstly, two radicals are coupled forming a dimer, and the process progresses with the coupling of monomeric radicals with dimer, trimers, and oligomers resulting in a complex branched polymer (lignification) [11,12].

The great complexity of the lignin structure is due to the variability of the linkages found in it, such as ether, esters, and carbon-carbon, the most abundant being the bond β-O-4, β-β, and β-5, as seen in Figure 2 [6,13].

Lignin has a great variability of functional groups in its complex structure. The main ones are shown in Figure 3. Their abundance in the structure also depends on the source of lignin [13,14,15]. The frequency and type of the most common linkages in softwood and hardwood lignins are described in Table 2.

## 2. Technical Lignins

Technical lignins are mostly obtained as co-products of the manufacture of cellulose pulp for paper, through wood pulping (delignification) processes [16,17]. The main objective of wood pulping is to liberate the cellulose fibers from the lignin binder and other non-fibrous compounds. The most common commercial processes can be grouped into four types: chemical, semichemical, chemimechanical, and mechanical [18].

In chemical pulping, delignification occurs until most of the lignin in the middle lamella of the woody cell is removed. This results in an easy separation of the fibers. Most of these processes are currently based predominantly on the sulfate (kraft) process and, less frequently, on the sulphite process [18].

On the other hand, in semichemical processes only a partial dissolution of lignin is achieved, as the wood chips are cooked during shorter periods of time or under milder conditions. Chemimechanical pulps are produced by pretreating the wood chips usually at elevated temperatures in alkaline solutions of sodium sulfite before defibration [18]. Lastly, mechanical pulping uses no chemicals, only mechanical abrasion combined with water or steam. Thermomechanical pulping has become the most important method of this kind. In this procedure, the refiners are first pressurized with steam at high temperatures in order to promote fiber liberation and operate at ambient temperature in a second stage [19].

During pulping lignin’s structure is inevitably modified. Therefore, the type of pulping process determines the type of lignin that is industrially obtained. Some examples of chemical pulping that will be addressed in this study are sulphite (lignosulphonates), kraft, soda, and organosolv lignins [6,16].

Although there is a great diversity of technical lignins, this article will focus mainly on lignosulphonates.

### 2.1. Lignosulphonates

Most of the pulp in the world was produced through the sulphite process until the 1950s. After that, the kraft process has been the dominating method. However, the sulphite process is still important in some countries and for certain pulp qualities [18].

Currently, the total annual global production of lignosulphonates is approximately 1.8 million tons. These technical lignins account for 90% of the total market of commercial lignin since the kraft process yields relatively small quantities of usable lignin [3,16].

This consists in the digestion of wood at 130–180 °C with an aqueous solution of a sulphite or bisulphite salt of sodium, ammonium, magnesium, or calcium [16,18]. There are several modifications of the sulphite method, which are designated according to the pH of the cooking liquor, as shown in Table 3 [18].

During the digestion process the linkages between lignin and carbohydrates are cleaved, as well as the carbon–oxygen bonds that connect lignin units. However, the sulphonation of the lignin aliphatic chain is the most important reaction as it lends the resulting LS their main characteristics [16].

This reaction consists in the attack on lignin’s structure by the negatively charged sulphite or bisulphite ions. The targets of this nucleophilic attack depend on the pH of the sulphite delignification. At high pH, quinone methide structures of phenolic units are the targets. However, at low pH, carbenium (benzylium) ions of phenolic or non-phenolic units are sulphonated at the position of the side chain, as shown in Figure 4 [16]. 

During pulping, about 4–8% sulphur is incorporated into lignin’s structure, making lignin water-soluble and preventing its recondensation, which would result in the redeposition of lignin on the cellulose fibers [16,20]. 

The resulting fiber pulp is separated from the spent pulping liquor through filtration and washing. This liquor contains 50 to 80 wt% of lignosulphonates, as well as hemicelluloses and residual pulping chemicals. The structures of these lignosulphonates may vary significantly due to the wide range of conditions under which sulphite pulping can be operated [3,16].

Purified forms of lignosulphonates are more valuable and have broader applications than crude spent sulphite liquor. The purification process can be performed according to a variety of techniques [5,16]. 

Ultrafiltration has been applied on an industrial scale to recover lignosulphonates from spent sulphite liquors. The higher molecular weight of lignosulphonates in comparison to other components in the spent liquors allows this method to be applied effectively. The result is a sugar-rich permeate and a retentate with up to 95% of lignosulphonates [3,5,16].

Alternatively, the spent sulphite liquor can be purified through alcoholic fermentation of the sugars. Lignosulphonates exhibiting a purity over 90% can then be recovered resorting to ultrafiltration [3,5,16].

The cation (or base) originally present in the pulping liquor can influence the physicochemical properties of lignosulphonates. For example, sodium sulphite produces longer lignin chains that are more suitable as dispersants. However, the use of calcium sulphite results in a more compact lignin [5]. Changing this cation may therefore be required for certain applications and can be achieved through the use of ion-exchange resins [16].

Lignosulphonates are soluble in water, have high molecular weights with a broad distribution, and a high ash content. These compounds also contain a variety of functional groups including phenolic hydroxyl and carboxylic groups and sulphur containing groups [20].

### 2.2. Kraft Lignin

The kraft process is the leading pulping process worldwide. Kraft pulp mills have a highly engineered incorporated system for recovery of pulping chemicals and energy. This system is crucial for the economic and environmental performance of these mills and relies on the combustion of the black liquor. Thus, the quantity of kraft lignin recovered for chemical use is low in comparison to lignosulphonates [12,16].

Kraft pulping uses a solution composed of sodium hydroxide and sodium sulphide, named white liquor, to cleave lignin’s ether bonds [5,18]. This process has a total duration of approximately 2 h and the temperatures range from 150 to 170 °C [21].

Kraft lignin is hydrophobic, highly modified, and displays a lower molecular mass than native lignin [22]. It is also not soluble in water and mostly solvent insoluble except for in highly alkaline mediums (pH > 11) [23].

### 2.3. Soda Lignin

Soda pulping was introduced in the 1850s as the first chemical pulping method [5,16,18]. Kraft pulping originated from this process and has almost completely replaced it, due to having better delignification selectivity which results in pulp with higher quality [18]. However, the soda process is currently becoming the main chemical pulping method of non-wood fibers such as bagasse, wheat straw, hemp, flax, kenaf, and sisal [5,16,20]. 

The soda process differs from the kraft process mainly in the cooking liquor that is sulphur-free [16,20]. Both kraft lignin and lignosulphonates can be classified as sulphur containing lignins, which are associated with environmental concerns [23]. Soda lignin, on the other hand, is sulphur-free, meaning that its chemical composition is closer to that of native lignin [20,24]. Soda lignins also differ from lignosulphonates in their low molecular weight, low levels of sugar and ash contaminants, and in being water insoluble. Thus, they are more similar to kraft lignins [16].

Soda lignins from non-wood plants differ from wood lignins since they contain more *p*-hydroxyl units, as well as high silicate and nitrogen [16,20].

### 2.4. Organosolv Lignin

With the environmental concerns associated with sulphur, many sulphur-free extraction processes have been developed, including the organosolv process [23]. Thus, lignin is extracted from plant-tissues with an aqueous solution of organic solvents at high temperature and pressure [12,22]. These solvents include methanol, ethanol, acetic acid, butanol, phenol, peroxiorganic acids, ethyl acetate, and formic acid [5,12,20,23]. 

The main advantages of this type of lignin is the absence of sulphur, as well as a less modified and more hydrophobic structure than kraft lignin, combined with lower ash content, higher purity, and usually lower molecular weight [5]. Organosolv lignins also contain many reactive side chains available for further chemical reactions [20]. However, this delignification process is not used widely, since it is expensive, requires a high amount of organic solvents, produces pulp fiber with low quality, and causes extensive corrosion of the plant equipment [5,12,23].

### 2.5. Comparison of Technical Lignins

As discussed, the lignin’s final structure and composition is highly influenced by its origin and extraction method [23]. Therefore, technical lignins display a wide range of properties [20].

Applications of these compounds require them to possess tailored properties. These include molecular weight, purity, homogeneity, and the presence of certain functional groups. The main chemical properties of technical lignins are summarized in Table 4 [20].

It is important to note that the information collected in Table 4 includes lignin from different sources. As seen in the table, lignosulphonates have the highest ash, sulphur content, molecular weight, and polydispersity [20].

Studies have shown that the molecular weight of LS also depends on wood species. Thus, hardwood LS displayed a lower weight than softwood LS [25].

All the lignins described in Table 4 contain a low amount of β-O-4 linkages (below 10%) and high amounts of C-C bonds, unlike native lignin. However, LS are water-soluble for most of the pH-range, which is not the case for kraft lignin [23].

## 3. Physico-Chemical Characterization of Lignosulphonates

As previously mentioned, the chemical structure and composition of lignin varies with the source and isolation method. Therefore, it is important to carry out a detailed characterization before it can be included in the production of value-added products [26]. However, this is not an easy task due to lignin’s three-dimensional architecture combined with a diversity of chemical linkages and functional groups [27]. 

Currently there are no uniform or standardized methods for the characterization of lignin. Nevertheless, efforts are being made to develop a series of ISO methods for the characterization of the following lignin features: general composition; functional groups; size and morphology; thermal properties; structural features; and safe handling and processability [28].

Thus, there is little information on fundamental analysis and characterization of technical lignins in literature, especially when it comes to lignosulphonates. This may be due to the declining use of sulphite pulping, the greater complexity of lignosulphonates, the lack of widely accepted standard protocols for the purification of these type of lignins, as well as the fact that the existing methods need to be adapted before they can be applied to lignosulphonates [16]. 

### 3.1. Chemical Composition

#### 3.1.1. Lignosulphonate Content

Measuring the amount of lignin in biomass is one of the most important steps in developing lignocellulosic biomass for bio-based chemicals. However, while attempting to achieve this task researchers have been presented with several challenges [29].

One of the most used wet chemical techniques to measure lignin content is the Klason or acid-insoluble lignin method. In this analysis the insoluble residues, after hydrolysis with H_2_SO_4_ 72% (*w*/*w*), are filtered, dried, and weighed. This acid-insoluble lignin is referred to as “Klason lignin”. A small portion of lignin is dissolved during hydrolysis. The remainder acid-soluble lignin is determined through ultraviolet (UV) spectroscopy, from the absorbance at 205 nm of the filtrated solution. The total lignin is determined as the sum of these two fractions. The Klason method is described thoroughly in ISO 21,436 [29,30]. Although this process is well established, the sample preparation and analysis are time consuming, costly and may destroy the sample [29]. This method may also be inapplicable when the biomass undergoes multiple transformations and therefore, unable to correctly determine lignin content. This is the case for all processes in which covalent bonds between lignin and secondary components, such as carbohydrates and their condensation products, are formed [31].

For commercial LS samples and spent sulphite liquors, LS content is frequently determined through the direct measurement of absorbance at selected wavelengths using UV spectroscopy. However, determining the extinction coefficient is not a simple task [29]. It should also be noted the wavelength choice may influence the obtained results. Measurements at 205 and 280 nm can be influenced by other substances, such as SO_2_ and carbohydrate degradation products, like furfural which also absorbs strongly at 280 nm [32,33]. Solvents for UV measurements of LS include water, and an ethanol/water mixture (2:8 *v*/*v*). Lin summarizes several absorptivity values at 280 nm found in literature for several types of lignin, including lignosulphonates [33]. Some studies also resort to the absorptivity value at 205 nm for acid-soluble lignin described in TAPPI UM 250 and ISO 21,436 [30,34]. Alonso and co-workers, on the other hand, determined the purity of commercial LS samples at 232.5 nm [35]. A procedure for the determination of lignin through the UV method is described by Lin [33].

Lignosulphonate content may also be determined through the Pearl-Benson method which is based on a chemical reaction involving nitrosation. Thus, the phenolic units in lignin react with acidified sodium nitrite forming a nitrosophenol. Then, upon addition of alkali, an intensely colored quinone mono-oxime structure is formed whose absorbance is measured at 430 nm and related to lignin concentration. However, a previous calibration with standard lignin is needed. This procedure is described in detail by Dence [32]. 

Near-infrared spectroscopy (NIR) has been used to successfully measure lignin content in samples. Firstly, a set reference samples must be prepared and the lignin content of these samples must be determined through a reference method. These data can then be used to calibrate the NIR signal. Thus, the lignin content in unknown samples can be rapidly quantified resorting to their NIR spectra [9].

#### 3.1.2. Total Ash Content

For polymer applications, lignosulphonates with low ash content are desired [36]. Therefore, it is important to determine the total ash content in samples. This parameter can be determined gravimetrically after incineration at 525 °C according to the procedure described in ISO 1762 (2019). In this method the samples are weighed in a heat-resistant crucible and ignited in a muffle furnace at 525 ± 25 °C. Then, the ash content is determined on a dry basis, using the mass of residue after ignition and the dry matter content of the sample [37,38].

### 3.2. Molecular Weight

#### Gel Permeation Chromatography (GPC)

Molecular weight (MW) is an important property of a polymer, that can provide information, such as degree of polymerization, even before investigating the chemical structure [26,39]. Gel permeation chromatography (GPC) or size exclusion chromatography (SEC) is the method of choice for the determination of the molecular weight distribution (MWD) of technical lignins [39,40,41]. However, it has been noted in several studies that secondary separation effects in GPC columns can interfere significantly with the obtained results [42]. 

These effects are caused by interactions between lignin molecules (association), lignin and solvent (solvatation), and lignin column packing material (adsorption), which should all be minimized in order to obtain absolute MWDs. Additionally, studies have found extremely high inter-laboratory deviations of calculated molar masses from similar lignin samples [42]. 

This method requires a previously made calibration from a series of standards of different MW, that display the relationship between molecular weight and elution time [39]. After calibration, the result can be converted into molar mass distribution. Studies have performed this calibration using polymers such as polystyrene sulfonates, pullulans, or proteins with mixed results. Thus, if the standards present do not present a similar molecular configuration to lignosulphonates, very large calculation errors may occur. Promising results have been obtained using lignosulphonate fractions with molecular weights determined previously by analytical ultracentrifuge for calibration [43].

Tetrahydrofuran (THF) is a widely used eluent, but most lignins are not entirely soluble in this compound. Therefore, a previous acetylation and/or methylation step is required [39,42]. Studies have attempted to acetylate lignosulphonates. However, these were only soluble up to 4.5% in the acetylation mixture. Even after this process, only the low MW compounds were soluble in THF. Therefore, it was concluded that the THF system is not suitable for lignosulphonates [44].

Various GPC systems have been applied for lignosulphonates and spent sulphite liquors analysis. Some are reviewed in Table 5 [39,42]. Some studies have achieved accurate results with in-line multi-angle laser light scattering (MALLS). This system solved previous problems caused by unrepresentative calibration standards, variations in refractive index, and fluorescence [45]. The complex mobile phase used by Fredheim and co-workers was chosen in order to prevent formation of aggregates and absorption to the column material [43,46]. 

### 3.3. Chemical Structure Characterization

Methods used for the characterization of lignin’s structure include spectroscopic methods such as Fourier transform infrared (FTIR), UV/vis and Nuclear Magnetic Resonance (NMR), and Raman spectroscopy [51]. Wet chemistry methods may also be used [9].

#### 3.3.1. Ultraviolet (UV) Spectroscopy

Ultraviolet (UV) spectroscopy refers to absorption spectroscopy in the UV region (200–400 nm). This method is one the most useful for the quantitative and qualitative analyses of lignin in solution. Because of its aromatic nature, lignin strongly absorbs UV light and exhibits characteristic maxima in the ultraviolet light region. Samples are most often liquids, thus lignin is typically dissolved in a solvent. Potential solvents for lignosulphonates include water and an ethanol/water mixture (2:8 *v*/*v*) [33,51].

The type of lignin, its chemical modifications, and the solvent used determine the location and intensity of the maxima. The spectrum of softwood lignin presents a maximum of absorbance at 280 nm, a shoulder at 230 nm, and a sharp peak at 200–210 nm. These bands are designated B, E_2_, and E_1_, respectively. On the other hand, the spectra of hardwood lignins exhibits B bands in the 268–277 nm range. Additionally, the B band absorptivity values for softwood lignins (18–21 L g^−1^ cm^−1^) are significantly higher than those for hardwood lignins (12–14 L g^−1^ cm^−1^). Lignosulphonates also present a lower light absorptivity when compared to kraft lignin, as sulphite pulping has a bleaching effect on lignin [33]. An example of an UV spectra of hardwood magnesium-based thick spent sulphite liquor (HLS) and softwood sodium lignosulfonates (SLS) in water is shown in Figure 5.

#### 3.3.2. FTIR Spectroscopy

Infrared (IR) spectroscopy has been used to characterize lignin since the early 1950s. This is due to the simplicity of the technique along with the fact that the samples do not need to be dissolved in any solvent and only small quantities are needed. Fourier transform infrared (FTIR) spectrometers are the most used equipment [52,53].

The mid-IR region is normally used to characterize lignin (4000–400 cm^−1^) [51]. Lignin IR spectra are easy to obtain. If possible, the preferred method for spectra acquisition should the KBr pellet transmission technique. Reflection techniques such as attenuated total reflectance (ATR) or diffuse reflectance (DR), as well as photo acoustic (PA) methods, should be used for analyzing liquids or high consistency pastes, and for surface analysis in the solid state [52].

IR spectroscopy is mostly used for the analysis of lignin in solid state. For liquid samples, UV and NMR spectroscopy are more frequently used. However, the need to measure IR spectra in solution exists when there is no time to isolate lignin from a solution. An example is the monitoring of spent pulping liquors for process control, which must be performed rapidly [52].

Due to the differences between the spectra of softwood and hardwood lignins, studies have used the IR spectrum as an indicator of the ratio of syringyl to guaiacyl units in lignins [53].

An example of a FTIR spectra of hardwood magnesium-based thick spent sulphite liquor and softwood sodium lignosulfonates is shown in Figure 6. The assignments for the main bands of milled-wood lignins have been explained in detail by Faix. Hemmilä and co-workers have also characterized two samples of ammonium and sodium lignosulphonates through FTIR spectroscopy and summarized their main absorption bands [36].

Advantages of FTIR spectroscopy include high signal to-noise ratio and linearity, high accuracy in frequency, the mechanical simplicity the spectrometer, and the easy data treatment due to well developed software for spectral data manipulation [52].

#### 3.3.3. Raman Spectroscopy

Much like in IR spectroscopy, in Raman spectroscopy an incident IR beam makes contact with the sample. However, in Raman spectroscopy, the photons involved are neither absorbed nor emitted but instead shifted in frequency. This shift is equal to the energy of the vibrational transition [54]. Raman spectra are determined through the difference between the frequency of the incident and refracted light. Each variation is related to one of the normal vibration modes of the molecules [55].

Raman spectroscopy and IR spectroscopy both provide information on the vibration and rotations of molecules and, in some cases, electronic transitions. Although the mechanism of these methods is different, the obtained information is complementary. The active vibrations in Raman may not be active in IR and vice versa. It should also be noted that in Raman spectra each band has a characteristic polarization which is very useful for the characterization of the molecular structure of a compound [55].

The first spectra of lignin samples revealed that the Raman signal was greatly obscured by laser-induced fluorescence. This problem was solved for most samples when a new Raman instrument based on NIR excitation was created. Thus, NIR Raman spectroscopy or NIR FT-Raman is rapidly becoming a promising technique in the analysis of lignin. However, the study of commercial lignins remains a challenge, as these samples produce a significant amount of fluorescence even when excited at 1064 nm. Therefore, further research is needed in other to develop more adequate approaches [56].

Advantages of the Raman spectroscopy include the ability to choose frequencies that allow selective lignin excitation and the easy characterization of heterogeneous samples. The latter case is a difficult task in IR spectroscopy due to the Rayleigh scatter of infrared photons [54]. 

Agarwal and Atalla summarized the main bands in the FT-Raman spectra of softwood and hardwood milled-wood lignins [56]. Ertani and co-workers also summarized the observed main absorption bands of the samples of two specialty lignosulphonates [57].

#### 3.3.4. Nuclear Magnetic Resonance (NMR) Spectroscopy

Nuclear Magnetic Resonance (NMR) spectroscopy is widely used for the characterization, classification, and detailed structural analysis of lignin because it provides information that cannot be obtained by chemical analysis. Nuclei of interest in NMR spectroscopy include: ^1^H, ^13^C, ^31^P. The most abundant carbon isotope, ^12^C, is not NMR-active, and the ^13^C isotope is only present in 1.11% relative abundance [58]. 

In an early stage, ^1^H NMR was frequently applied for lignin analysis due to fact that the proton nucleus is of 100% natural abundance and of high sensitivity. This technique is used almost entirely on acetylated lignins as this provides better signal resolution. ^1^H NMR is able to quantify a number of notable lignin structural features. Nevertheless, it has drawbacks such as a limited range of chemical shifts, extensive signal overlapping, and proton-coupling effects [53].

^13^C NMR has contributed greatly to the current knowledge of lignin’s structure. However, this technique has an even lower sensitivity than ^1^H NMR, due to the even closer energy levels than and the low natural abundance of ^13^C. To obtain good spectra in acceptable times sample sizes in the milligram range are required. Despite apparent sensitivity issues, NMR spectra contain more information than those of other spectral techniques. NMR alone can often fully identify compounds as well as determine their structure and bonding patterns, even in complex molecules [58]. 

Other approaches with increased sensitivity have been developed [58]. In two-dimensional (2D) NMR, particularly heteronuclear single quantum coherence (HSQC) NMR, a ^13^C-^1^H correlation spectrum is obtained where each signal corresponds to a unique C–H bond in the analyzed sample. Currently, this technique is commonly used for the elucidation of lignin structure and bonding [4,53]. This approach leads to a much easier analysis since the overlapping of peaks is completely avoided or largely minimized [4].

Unlike other types of “non-polar” or “non-sulfonated” technical lignins, LS have not been studied extensively by NMR spectroscopy. A major cause for this is that LS are insoluble in almost all organic solvents. LS contain some paramagnetic ions which cause signal broadening due to an increased relaxation rate. Another issue is that LS are usually used as metal salts. When these are dissolved in water the currents in the conducting solution and sample heating can difficult NMR analysis [43].

The methods present in literature for making lignin soluble in organic solvents are not applicable to LS. However, when LS is ion exchanged to the acid form it becomes soluble in methanol. This treatment also removes most of the metal ions from the solution, thus eliminating previously mentioned difficulties. Using methanol-d4 instead of D_2_O improves spectra resolution [43].

Lebo and co-workers successfully studied eight different commercial LS samples using this method. Thus, important structural information was obtained [43]. 

On the other hand, Marques and co-workers studied purified LS from thin and thick spent liquors by 1D/2D NMR, using deuterated water with Sodium 3-(trimethylsilyl)propionate as an internal standard. In this study, the major obtained carbon signals in the ^13^C NMR spectra of LS were summarized [59].

#### 3.3.5. Wet Chemistry Methods

Wet chemistry methods have also been extensively used to characterize the structures of different lignins. Contrary to the spectroscopy methods previously described, these methods degrade the sample and the information is obtained through the resulting products [9]. Chemical degradation reactions of lignins were the only available techniques for structural characterization before the appearance of NMR methods [60]. 

Frequently used wet chemical techniques used for quantifying lignin monomers are acidolysis, cupric oxide, NBO, permanganate oxidation, and thioacidolysis [29]. Each method can provide a piece of structural information but in most cases, the information is semi-quantitative at best [9]. Therefore, these methods will not be addressed further in this study.

### 3.4. Functional Group Analysis

#### 3.4.1. Determination of Phenolic Hydroxyl Groups

One the most important factors which influences the physical and chemical properties of lignin are the phenolic hydroxyl groups. The chemical reactivity of lignin is deeply affected by its phenolic hydroxyl content, namely in the reaction with formaldehyde for the production of adhesives. Therefore, determining the content of these groups provides insights on the structure and reactivity of lignin samples [61].

Methods that are frequently used to measure phenolic hydroxyl content in lignin include potentiometric and conductometric titration, ionization UV spectroscopy, and NMR spectroscopy [61]. However, when it comes to lignosulphonates, this quantification has been challenging for researchers [62].

The UV method is quick, simple, and applicable to lignosulphonates [62]. In alkaline solutions, phenolic hydroxyl groups are ionized and the absorption shifts towards longer wavelengths and higher intensities. Therefore, it is useful to examine the ionization spectrum (∆ε_i_), that is determined though the subtraction of the neutral solution spectrum by the spectrum derived from an alkaline solution. Thus, the quantification of the phenolic hydroxyl groups is based on this principle [51]. Lin provided a procedure for this method in detail [33]. However, lignin model compounds or a lignin of known phenolic hydroxyl content need to be used for calibration. This may introduce errors in the obtained values since the structure of these model compounds can be simple in comparison to technical lignins [61,62]. 

Alonso and co-workers determined the content of phenolic hydroxyl groups in lignosulphonates by the Goldschmid method (UV absorption). The spectra of an alkaline lignosulphonate solution (boric acid pH = 12) were measured against a neutral lignosulphonate solution (potassium dihydrogen phosphate pH = 6). The phenolic hydroxyl content was calculated from the absorptivity at the maximum of 250 and 400 nm [63,64].

The titration method can be used to quantify this group along with carboxyl or sulfonate groups. This procedure is based on the acidity of phenolic hydroxyl groups and relies on an internal standard. However, this method does not provide information on the distribution of the different types of phenolic OH groups. The obtained results may also be influenced by other substances such as methoxyl groups, high sugar contents, and sulphites [61,62].

When this analysis was attempted via ^1^H or ^13^C NMR spectroscopy, a previous acetylation step was required. However, as previously mentioned, this reaction could not be completed with lignosulphonates, resulting in inadequate results [62].

^31^P NMR has also been used to quantify hydroxyl and phenolic groups in lignin, but a functionalization step where these groups are phosphitylated is still required [4]. Stücker and co-workers analyzed several softwood lignosulphonates using this spectroscopic method. Firstly, the LS samples were converted into their corresponding lignosulphonic acid by ion exchange treatment and later lyophilized. These samples were then dissolved in a mixture of anhydrous N,N-dimethylformamide (DMF), deuterated DMF and pyridine. Endo-N-hydroxy-5-norbornene-2,3-dicarboxylic acid imide was used as an internal standard. The relaxation agent and phosphitylation reagent were chromium (III) acetylacetonate and 2-chloro-4,4,5,5-tetramethyl-1,3,2-dioxaphospholane, respectively. The authors concluded that the proposed method was accurate for the determination of hydroxyl groups in LS samples [62].

#### 3.4.2. Determination of Methoxyl Groups

The methoxyl content of lignin is frequently determined according to Zeisel procedures. ^13^C NMR spectroscopy has also been used as well as on ^1^H NMR, although in the latter case for a rough estimation based on spectral examinations [65]. 

The original Zeisel procedure cannot be used to determine methoxyl groups in sulfur containing compounds. Therefore, it is not suited for lignosulphonates. In order to apply this method these compounds, modifications were proposed by Vieböch and co-workers. Through this method, it is possible to express the methoxyl content of lignin on a particular basis, such as equivalents per C_9_ unit but, lignin must be previously purified. If the Klason lignin content of the sample inferior to 95%, further purification should be carried out. Chen and co-workers have explained this method in detail [66].

Marques and co-workers determined the content of methoxyl groups in purified and unpurified sulphite liquor using the Zeisel-Vieböch-Schwappach method and ^13^C NMR. For the former method, 0.5 g of phenol, 1 g of KI, and 2 mL of H_3_PO_4_ (85%) were added sequentially to 20 mg of the samples. The obtained mixture was then heated to 145 °C for 40 min and then dragged to a solution of Br_2_ by a current of N_2_. Next, the Br_2_ was dragged with a solution of sodium acetate (20%) and destroyed with formic acid (4%). Then, 20 mL of H_2_SO_4_ (10%) and 5 mL of KI 10% were added. Lastly the mixture was titrated with Na_2_S_2_O_3_ 0.1 M until it turned pink. Starch was used as an indicator. The quantification through the ^13^C NMR method was achieved through the integral of the centered signal at 56.8 ppm. Although different results were obtained with these two methods, this difference was not considered to be significant [13].

More recently, another method has been proposed by Sumerskii and co-workers with higher precision, accuracy, and sample throughput than the Zeisel-Vieböch-Schwappach method. This new approach is based on headspace-isotope dilution Gas chromatography–mass spectrometry and can be applied to any type of lignin [31]. 

### 3.5. Thermal Properties

Thermogravimetry (TGA) and differential scanning calorimetry (DSC) are the most frequently used techniques in lignin chemistry for determining its thermal properties. Thermal analysis can be applied to study the physical properties of lignin as well as its derivatives. In such studies, it provides information on molecular arrangements, phase transitions, and the interaction between lignin and low molecular weight substances such as water. For more specific applications, these techniques may be used determine the durability of lignin and its respective glass transition and degradation temperature. A great advantage is that only a small amount of the sample is used and any kind of material may be analyzed, including powders and liquids [67].

DSC is often used to determine the glass transition temperature and heat capacity. Variations in these values can be induced by low molecular weight contaminants, molecular weight, thermal history, cross-linking, and pressure. Therefore, thus, a glass transition temperature value for any specific lignin type cannot be precisely reported [51,67]. 

TGA is primarily conducted dynamically and provides information about degradation temperatures and other phase transition temperatures [51].

A disadvantage of thermal analysis is that although it may provide accurate data on transition and degradation temperature, it only gives general information on the molecular arrangements of polymers can be obtained through these techniques [67].

## 4. Approaches to Increase Reactivity

Lignin’s guaiacyl (G) units and phenol present a similar structure, as shown in Figure 7. However, in the aromatic ring of lignin’s G units positions 1 and 3 are blocked. On the other hand, in the aromatic ring of lignin’s syringyl (S) all of the ortho and para positions are blocked. Therefore, the methoxyl groups are the cause of lignin’s low reactivity. This factor, alongside its high dispersity in molecular weight, has hindered the application of industrial lignins in resin synthesis [23,68].

However, it should be noted that, in recent studies, it was suggested that hardwood lignins, although containing a lower number of reactive sites in their aromatic rings, could still be used as a phenol substitute in phenol-formaldehyde (PF) resins. It was suggested that other factors may also influence the final performance of the obtained resin such as: molecular branching and conformity, steric availability, molar mass, and its distribution [69]. The amount of phenolic and aliphatic hydroxyl groups is also highly important for all adhesive applications that require reaction of lignin with different aldehydes, phenols, tannins, or isocyanates [70]. 

Several methods to improve the reactivity of lignin molecules have been reviewed in literature and will be discussed in this study.

### 4.1. Phenolation

Phenolation or phenolysis is one of the most promising treatments for lignin modification. This method allows the reduction of lignin’s molecular weight as well as an increase of its phenolic hydroxyl groups, therefore increasing its reactivity. Phenolation reactions result in the attachment of phenol onto lignin. These reactions can occur in alkaline or acidic mediums [71].

In the first step of the reaction, the protonation of the benzyl hydroxyl group occurs. The next step is the dehydration at the α-carbon, forming a carbonium ion. An electrophilic attack to the phenol molecule by the carbonium ion produces a phenol condensation product, adduct. Then occurs the incorporation of ortho or para-phenyl to the α-hydroxyl groups of the propane side chains of lignin. Lastly, adduct fragmentation takes place. This results in a decrease in the molecular weight of the reaction products, thus facilitating their incorporation in resins. Depending on the reaction conditions, side reactions may occur. Thus, a self-condensation product could be produced by the reaction of the carbonium ion in a lignosulphonate molecule. The treated lignin can react in an acidic or alkaline medium with formaldehyde to produce resol or novolac resins [72].

A recent study has suggested that lignin substructures such as β-O-4′, β-5′/α-O-4′, β-β’, and α-carbonyl react with phenol, thus increasing the amount of phenolic hydroxyl groups present in the structure. This work also considered the ortho and para positions for lignin phenolation and the presence of more substructures, shown in Figure 8, because of the elimination of formaldehyde from the γ-carbon [71].

Phenolation is frequently used for the modification of lignosulphonates since it increases the content of phenolic hydroxyl groups, while reducing molecular weight, thus simplifying their structure. The removal of the sulfonic acid group after phenolation has also been reported [71].

Alonso and co-workers studied the phenolation method to modify the structure of softwood ammonium lignosulphonates. The lignosulphonates reacted with phenol using oxalic acid as a catalyst. The methods used to characterize the final products were GPC, FTIR, and ^1^H NMR. It was concluded that high temperatures, a long reaction time, and low lignosulphonate concentration favored high phenol conversion rates. Therefore, the chosen ideal reaction conditions were 120 °C, 160 min, and 30% lignosulphonate content [73].

Hu and co-workers used phenolation to modify lignosulphonates for the synthesis of a phenolic resol with phenol and formaldehyde. For the phenolation process, 20 g of lignosulphonate were added to 100 g of molten phenol. The pH of the mixture was adjusted to be between 9 and 10 through the addition of sodium hydroxide aqueous solution. The resulting mixture was stirred and heated slowly to 100–120 °C in an oil bath, then refluxed for 1 h. The reaction was quenched by cooling to 70 °C [74].

A major advantage of phenolation is that the final product is soluble in phenol. Therefore, direct incorporation in PF resin production may be possible without prior purification [3].

### 4.2. Hydroxymethylation

Lignin methylolation, or hydroxymethylation, introduces hydroxymethyl (–CH_2_OH) onto lignin molecules. The obtained lignin can be directly incorporated in the synthesis of PF resol resins for wood adhesives as a phenol substitute. Lignin methylolation usually occurs in an alkaline medium with formaldehyde [72].

Three reactions occur. In the main reaction, the Lederer–Manasse reaction, lignin’s reactivity is increased through the incorporation of hydroxymethyl groups into the aromatic rings, as shown in Figure 9-reaction a. The Tollens reaction may also take place, but it is not desirable. This involves the substitution of lignin’s side chains by aliphatic methylol groups, Figure 9-reaction b [75]. With the increase in temperature, this reaction may be followed by a condensation in which hydroxymethyl groups react at free positions of other lignin units to form methylene bonds (Figure 9 reaction c), thus decreasing hydroxyl group content. Another undesirable side reaction is the Cannizzarro reaction, in which formaldehyde reacts with itself [35,72,75].

Lignin’s reactivity in hydroxymethylation or cross-linking depends on the reaction conditions but, also on lignin’s source and pulping process [75].

Alonso and co-workers studied the methylolation of softwood and hardwood LS. FTIR and ^1^H NMR were used to evaluate these samples before and after methylolation. Softwood ammonium LS were used to optimize the operation conditions to promote the Lederer–Manasse reaction, as they displayed the most promising characteristics. The reaction was followed by the changes in the concentration of free formaldehyde. The chosen optimum operating conditions for methylolation were 1.0 of formaldehyde-to-lignin molar ratio, 45 °C, and a sodium hydroxide-to-lignin molar ratio of 0.80. Hardwood LS exhibited lower reactivity due to their predominant structural unit, syringyl, having a higher degree of substitution on the aromatic ring [35].

It should be noted that the reaction of formaldehyde at the meta position of phenolic hydroxyl groups has been reported to occur when all the ortho and para positions are blocked, as in syringyl units. However, it is significantly slower and incapable of saturating all of the available meta positions [76].

Pang et al. used a different procedure for the methylolation of calcium LS. Initially, the solution of calcium LS (30%) was heated to 80 °C. The pH was adjusted to 11 with NaOH, under stirring. Then, a 37% formaldehyde solution was added slowly to the mixture until a mass ration of formaldehyde to lignosulphonate of 0.35. After a reaction time of 2 h, the process was ended [77].

Aro and Fatehi suggested the purification of LS after methylolation through drying or membrane filtration [3]. 

However, as formaldehyde is a known carcinogenic, studies have proposed the total substitution of formaldehyde (LD50 rat > 800 mg/kg) in lignin-based wood adhesives with glyoxal, a less toxic aldehyde (LD50 rat > 2960 mg/kg), which is non-volatile with regard to the aqueous phase in the lignin hydroxymethylation step [78,79,80].

Mansouri and co-workers proposed a procedure for the glyoxalation of LS for later incorporation in wood adhesives. Firstly, calcium LS powder (96% solid) was slowly added to water in the proportion of 29.5 to 38.4 parts by mass. Sodium hydroxide solution (30%) was added when necessary, in order to maintain the pH between 12 and 12.5. This step, alongside vigorous overhead stirring, allowed for adequate dissolution of LS. In total, 18.1 parts by mass of sodium hydroxide solution (30%) were added, resulting in a final pH of 12.5. The solution was then transferred to a 250 mL flat bottom flask equipped with a condenser, thermometer and magnetic stirrer bar and heated to 58 °C. Then, 17.15 parts of glyoxal (40% in water) were added. Lastly, the mixture was continuously stirred with a magnetic stirrer for 8 h [78]. 

### 4.3. Oxidation

Oxidative lignin conversion is a promising way to increase lignin’s value by converting it into chemicals. In this process, lignin’s structure is disassembled into its phenolic building blocks, which can be converted into targeted end products. This method can be used to produce highly functionalized, valuable structures such as vanillin and is, therefore, of great interest [81].

The mechanism of lignin depolymerization depends on the oxidants and catalysts used. The most used oxidants include oxygen, hydrogen peroxide, ozone, and peroxyacids of chlorine (including chlorine dioxide). The involved mechanisms are electrophilic, radical, and nucleophilic [82].

Oxygen is the most promising oxidant, both in terms operability and costs. When oxygen is used as an oxidant, radical and electrophilic mechanisms are responsible lignin’s degradation. The main end products are vanillaldehyde/acid, syringaldehyde/acid, and *p*-hydrobenzaldehyde/acid [82]. The structure of these aldehydes is shown in Figure 10. 

Lignin oxidative conversion follows three steps: side chain cleavage, ring opening, and condensation [82].

In case hydrogen peroxide is used, the mechanisms involved are of radical and nucleophilic nature, resulting in aldehydes, acids, and quinines as the main products [82]. 

The catalysts used in oxidative lignin conversion can generally be divided into three types: metal-inorganic catalysts, metal-organic catalysts, and organic catalysts. Metal oxide presents high activity, low cost, and is easy to manipulate; therefore, it is seen as one of the most promising catalysts for oxidative biomass conversion [82].

Santos et al. studied the oxidation kinetics of hardwood magnesium LS with oxygen under alkaline conditions. The reaction system O_2_/NaOH resulted mainly in the production of vanillin and syringic aldehyde. Studies with the addition of a catalyst (copper salt, 20% *w*/*w*) were also conducted resulting in an increase in yield of 25 to 50%. The highest obtained yields of syringic aldehyde (16.1%) and vanillin (4.5%) were obtained using the following reaction conditions: 150 °C for 20 min in 0.9 M alkaline solution and oxygen pressure of 10 bar. It was noted that the sugars present in the sulphite liquor significantly reduced the yield of the aromatic aldehydes. Thus, the removal of these compounds prior to LS oxidation was highly advised [83].

Yuan and co-workers oxidized ammonium LS with H_2_O_2_ for subsequent incorporation in a binder for a wood-based green composite. Firstly, 50 g of LS was mixed with 100 g of water. The pH of the resulting solution was adjusted to 10 using a sodium hydroxide solution (30% *w*/*w*). Next, a 30% H_2_O_2_ dosage based on the dry weight to LS was added. The mixture was heated to 60 °C and stirred for 30 min. Lasty, the solution was kept at 80 °C in an oven for 6 h and then cooled to ambient temperature [84].

Other procedures were suggested in different studies, namely for the incorporation of LS in a foaming resol resin as a partial phenol substitute [77,85].

Oxidation with hydrogen peroxide has been widely used in the pulping industry and is environmentally friendly. However, the decomposition of H_2_O_2_ forms molecular oxygen as well as several different radical species. These compounds react with lignin in a variety of ways, thus creating a complicated reaction mechanism [3].

### 4.4. Hydrolysis

In the hydrolysis process, water is used to cleave the ether bonds that connect lignin’s phenyl-propane units, usually in the presence of an acid or an alkaline catalyst. The final product has a lower molecular weight. In lignin’s case, hydrolysis can occur in an alkaline or acidic medium under sub or supercritical conditions [15,71].

In acidic hydrolysis, lignin is reacted with a strong acid and catalyst, resulting in the demethylation of the phenyl-propane unit and in an ortho substitution by hydroxyl groups. Hydrolyzed lignin is subjected to high temperatures in order to cleave the ether bonds of the phenyl-propane units in the lignin structure. This results in more reactive monomeric units [71].

In an alkaline medium, –OH groups act as a primary catalytic agent and are commonly used in pulping reactions for severing lignin. These groups attack different lignin bonds and produce ether bonds that can be cleaved easily through other treatments [71].

Whether it was subjected to an acidic or alkaline treatment, the treated lignin is susceptible to degradation, thus producing new phenolic hydroxyl-containing units with low molecular weight. In an alkaline treatment, the products of this degradation can be of value in condensation reactions to produce phenolic products. Hydrolysis in an acidic medium usually occurs in non-aqueous media; therefore, the obtained product must be previously extracted from an organic solvent before its use [71].

Mansouri and co-workers increased the reactivity of softwood LS using alkaline hydrolysis with sodium hydroxide. The modified samples were tested according to the content of phenolic hydroxyl groups, aromatic protons, as well as weight-average MW, number-average MW, and LS content. UV spectroscopy, ^1^H NMR, and aqueous GPC were used for this characterization. The powder LS was used with a ratio of 1/10 (*w*/*w*) to the sodium hydroxide solution, 2% (*w*/*w*). The optimum operating conditions were 170 °C and a reaction time of 90 min. The LS modified under these conditions presented a high concentration of phenolic hydroxyl groups, aromatic protons, as well as high LS content and low MW. The reactivity towards formaldehyde was also tested through a methylolation reaction. In this test, the modified LS displayed an increase in reactivity in more than 50% when compared to the original lignin. Thus, the authors concluded that increasing the severity of the operating conditions produced largely improved formaldehyde reactivity [86].

However, studies have indicated that lignin yields can decrease at extreme reaction conditions due to re-polymerization, especially under acidic conditions. This reaction can be controlled and the conversion rates can be increased resorting to alkaline catalysts. Catalysts can also increase selectivity by selectively cleaving bonds [15].

### 4.5. Ionic Liquid (ILs) Treatment 

Although glyoxilation, phenolation, and hydroxymethylation significantly influence the reactivity of lignin in resin synthesis, they rely on volatile organic reagents. Thus, there is still a need for more eco-friendly methods [3,68].

Ionic liquids (ILs) present chemical and thermal stability, low vapor pressure, and high ionic conductivity [68]. Their low melting point does not exceed 100 °C and, due to their negligible vapor pressure, these liquids obey the principles of green chemistry [87]. These properties have resulted in an increasing interest in these compounds in different sectors as “green solvents” [68].

In lignin chemistry, ILs can be modifiers or depolymerizing compounds [87]. One of the most recent methods to improve lignin’s reactivity is the pretreatment with ILs such as 1-Butyl-3-methylimidazolium chlorine ([Bmim]Cl) and 1-ethyl-3-methylimidazolium acetate([Emim][OAc]) [68].

These ILs were applied to treat industrial lignin by Qu and co-workers. In this study, ([Bmim]Cl) or ([Emim][OAc]) were mixed with industrial lignin at a ratio of 20:1, respectively. Then the mixture has heated at 120 °C for 30 min. In order to recover the lignin by vacuum filtration, deionized water was added at a ratio of 5:1. The recovered solid was washed repeatedly with deionized water until the filtrate was colorless. Lastly, the obtained lignin was dried overnight in a vacuum oven at 40 °C. Each treatment was carried out twice. This procedure reduced the average MW and polydispersity of the treated lignin. When [Emim][OAc] was used the phenolic hydroxyl and carboxyl groups were preserved but the methoxyl groups were reduced by 45% [88].

In a later study by Younesi-Kordkheili et al., soda bagasse lignin was modified through this method for incorporation in a UF resin. Modified lignin, when compared to unmodified lignin, had a better performance as an additive in these resins [68]. In a following study, Younesi-Kordkheili and co-workers proposed the mechanism shown in Figure 11 for the reaction between [Emim][OAc] and lignin [89].

In another study, ionic liquid lignins were prepared from sodium LS by a cation exchange to modify its thermophysical properties. When sodium was replaced by the tris-[2-(2-methoxyethoxy)ethyl]amine cation a flowable ionic liquid lignin was produced with a Tg of −13 °C. Through this approach, the properties of both LS and ILs were combined to create a dispersant and binder for cellulose and gluten bio composites. Panels with fewer defects and improved toughness were produced with modified LS in comparison to those with unmodified LS or free ionic liquid. The authors concluded that the retention of OH groups and the low Tg value of the modified LS was essential to produce composites resistant to high stress. Hence, this treatment introduced new functionalities while maintaining the properties of LS [90]. 

## 5. Lignosulphonates in Wood Adhesives

As previously mentioned, despite being produced in high quantities, lignin is usually burned for energy production [2,3,91]. 

Nevertheless, lignin displays several characteristics which make it a promising polymer for the development of new products. These characteristics include good rheological and viscoelastic properties, good film-forming ability, small particle size, and compatibility with a wide range of industrial chemicals. Depending on its origin, lignin can be of hydrophilic or hydrophobic nature allowing for the production of a wide range of blends. The presence of aromatic rings in its structure also provides thermal stability, good mechanical properties, and the possibility of a broad range of chemical transformations [5].

Lignosulphonates are currently used for a variety of value-added products, as shown in Figure 12; however, here we will focus only on wood adhesives [5].

Since the start of wood pulping, the large quantities of generated lignin waste have been proposed for the preparation of adhesives. Many studies have been conducted on the use of lignins to produce wood adhesives, including lignosulphonates [23,78].

The most successful industrial attempts to use lignin for wood adhesives consisted in using this polymer as a partial substitute in phenol-formaldehyde (PF) resins or urea-formaldehyde resins [23,78]. 

Nevertheless, these resins include formaldehyde which is a known carcinogenic. Nowadays, concerns about formaldehyde emissions and the sustainability of the used raw materials and final products are reducing the popularity of formaldehyde-based synthetic resins for wood-based panels [23,68,92]. Thus, formaldehyde-free systems which combine LS with polyethylenimine, furfuryl alcohol, polymeric isocyanate, chitosan, tannins, glyoxal, and wheat flour have been proposed [70,78,84,93,94,95].

### 5.1. Formaldehyde Adhesives

#### 5.1.1. Lignin-Urea-Formaldehyde (LUF) Resins

In the second half of the 20th century, a considerable number of studies were developed which report the incorporation of 5 to 20% of LS in UF resins, after or during its synthesis. The produced UF resins were reported by some works to display increased cold adhesion to wood particles and higher storage times. On the other hand, the resultant particleboards had improved mechanical properties, as well as lower formaldehyde emissions [92].

One of these studies added hardwood calcium-based spent sulphite liquor from the acid sulphite process (55–60% total solids, pH 3.0 to 3.5) during the UF synthesis. This type of LS was selected as calcium was suspected to catalyze the reaction. To produce the resin, acidic methylolation was accomplished with 3.0 to 5.0 parts of formaldehyde at 90 to 95 °C for 12 to 24 h. The acidic medium was preferred as it allowed the reaction to take place preferably at the meta positions, which are more readily available. Next, the temperature was decreased to 70–80 °C and 20 parts of urea were added and left to react for two hours. Then, 22.5 parts of formaldehyde were added for another reaction period of four to six hours at 70–85 °C. Lastly, 5.8 parts of ammonia were added providing a final pH of 8.0–8.5. The authors claimed that the obtained resin bonded wood particles when used at concentrations of 1 to 5% by weight while displaying a superior water resistance to the unmodified UF resin. It was concluded that all the urea reacted with the methylol lignosulfonate [96].

A recent study attempted to incorporate unmodified or hydroxymethylated thick spent sulphite liquor (TSSL) into a standard UF resin, with less promising results. TSSL was added in different amounts (10 and 20%) during the LUF resin synthesis alongside the second urea or after the synthesis. It was found that wood particleboards with 90% the amount of UF resin alone displayed similar IB strengths, as well as improved thickness swelling (TS) than those produced with any of the modified resins. The authors concluded that unmodified or hydroxymethylated TSSL acted as either inert or detrimental compounds in the performance of the UF resin, depending on the TSSL modification and incorporation stage. However, incorporation of unmodified or hydroxymethylated TSSL during the synthesis led to lower condensation times of the resins, lowering production costs [92]. 

These authors noted a higher acid buffer capacity of the LUF resins, which was more pronounced when LS was added during synthesis. Bekhta and coworkers also reported this effect. Thus, the sulfite and bisulfite ions in the sulphite liquor create a buffer system, therefore increasing the amount of acid needed to change the pH and slowing down resin cure [92,97]. This factor alongside the presence of ashes and sugars may significantly increase gel time [92]. 

Antov et al. combined an UF resin with commercial ammonium LS for the production of high-density fiberboard (HDF) panels from hardwood fibers. HDF panels were produced with low content UF, 3%, and ammonium LS content of 6, 8, and 10%, based on the dry fibers. The physical and mechanical properties of the fiberboards, such as water absorption (WA), thickness swelling, modulus of elasticity (MOE), bending strength (MOR), internal bond strength (IB), as well as formaldehyde content, were examined. The produced HDF panels complied with the requirements for use in loadbearing applications in humid conditions. The obtained formaldehyde content was also extremely low, and equivalent to that of natural wood. Notably, panels produced with more than 8% of ammonium LS displayed better physical and mechanical properties than the control panels with only UF resin (at 6%). To conclude, the authors suggested that future studies should add suitable cross-linking agents and study the bonding interaction of all the components in these panels, in order to decrease the hot-pressing factor [98].

It should be noted that the low formaldehyde release reported with the study while using ammonium LS may be caused by the reaction between the ammonium ion and formaldehyde under the conditions applied to the panels [76].

Gao and co-workers tried a different approach: incorporating sodium LS in UF resins as an anionic surfactant or stabilizer in order to improve colloidal stability and extend storage stability. When LS was added at the methylolation stage, the storage stability improved from 30 to 200 days. The nature of the interaction between LS and UF resins was studied using techniques such as FTIR, ^13^C cross polarized magic angle spinning NMR. It confirmed that LS could increase the electrostatic repulsion of UF resins thus avoiding aggregation and aging. However, no chemical reaction between UF resins and LS was observed. As previously noted in other studies, LS addition slightly reduced the curing rate; but the thermal stability of the UF resin was improved. The shear strength and formaldehyde release of the obtained resins were also comparable to standard UF resins [99].

#### 5.1.2. Lignin-Phenol-Formaldehyde (LPF) Resins

Ghorbani and co-workers studied four commercial spruce LS for different sulphite pulping processes as partial (40% *w*/*w*) phenol in lignin-phenol-formaldehyde (LPF) resole resins. The incorporation of the studied LS resulted in a faster viscosity gain during resole cooking compared to the lignin-free reference resin. Although displaying the lowest average MW and dispersity, sodium LS, when incorporated into the LPF resin for gluing beech veneer strips, resulted in the best curing and tensile shear strength development under hot pressing. On the other hand, calcium and magnesium LS were less suited as phenol replacements, since the obtained LPF adhesives had a poor performance. In this study, phenolation of sodium and ammonium LS, which had the most promising characteristics, did not significantly improve the performance of the obtained LPF resins [100].

Studies have also shown that impurities in crude lignin, specifically elemental sugars, lowered the reactivity of lignin towards formaldehyde for the incorporation in LPF resins, thus leading to extended pressing times. The strength and water resistance of the obtained resins were also hindered [101,102]. 

Alonso and co-workers used methylolated softwood ammonium LS as a phenol substitute in the synthesis of a LPF resin. The optimum operating conditions for this synthesis were studied. The obtained LPF resins were characterized according to free phenol and formaldehyde contents, gel time, alkaline number, viscosity, pH, solid content, and chemical structure changes (^13^C NMR). The optimum operating conditions were sodium hydroxide to phenol-modified LS molar ratio of 0.6; formaldehyde to phenol-modified LS molar ratio of 2.5; and 35% *w*/*w* of replaced phenol by LS. The obtained LPF resin under these conditions complied with the specifications necessary for its utilization in plywood. Its characteristics were also similar to those of the commercial PF resol resin used as a reference [63].

In a later study, the structural (FTIR), thermal (TGA), and rheological properties of a LPF resin 30 wt% of replaced phenol by methylolated softwood ammonium LS were studied and compared to those of a commercial PF resin. While the PF resin displayed a higher reactivity, the TG and DTG thermograms revealed that the addition of LS improved the thermal decomposition temperature of the LPF resin. The DSC profile of these resins was similar, indicating that LS functioned as an extender. Lastly, it was concluded that the flow behavior of both resins was different, as PF exhibited a behavior close to a Newtonian flow, while the LPF resin displayed a pseudoplastic behavior. This was explained by the increased branching obtained for the LPF resin. The authors concluded that this change in behavior may be beneficial for certain applications and suggests that a better mechanical performance can be obtained for LPF resins [103].

Antov et al. assessed the viability of calcium LS as an eco-friendly additive in a PF resin, at different ratios, for medium-density fiberboards (MDF). It was concluded that addition of LS should not exceed 10% (on the dry fibers) to avoid deterioration in the mechanical properties. The authors recommended a content of 3.5% PF resin to produce LS-PF bonded MDF panels that obey EN standard requirements [104].

### 5.2. Formaldehyde-Free Adhesives

Spent sulphite liquor alone has been used as a wood adhesive; however, it was associated with high pressing temperatures and long pressing times. Combinations with strong mineral acids and hydrogen peroxide have been proposed; however, neither have found industrial success [76].

Unmodified magnesium LS alone has been studied as a binder for industrial waste fibers by Antov and co-workers. Composites were produced using a hot press temperature of 210 °C, a pressing time of 16 min, and a 15% gluing content of magnesium LS (based on the dry fibers). The obtained MOR and MOE values were higher than the minimum required for PBs for interior fitments for use in dry conditions and equivalent to the value for MDF panels. However, these composites showed deteriorated moisture properties [101]. 

Ferreira et al. also confirmed the adhesive properties of TSSL alone, allowing the production of PBs with IB values above the requirements of standard EN 312 for general purpose boards for use in dry conditions (type P1). The improvement of PBs performance was achieved by adding wheat flour (WF) to TSSL, particularly when the mixture was pre-heated at 94 °C [95].

In a later study, manufacturing conditions for the particleboards using TSSL and WF were investigated. It was possible to produce PBs with densities ranging from 682 kg m^−3^ to 783 kg m^−3^, temperatures from 180 to 210 °C, and pressing times between 8 to 10 min. All the particleboards produced in these conditions were in accordance with the IB requirements of standard EN 312 for PBs type P2. These standards were also obeyed when the PBs were manufactured 20 days after binder preparation. However, the authors concluded that further studies need to be conducted to improve water resistance and decrease the pressing time, allowing the increase in the range of application of these PBs [91]. 

In other studies, the use of unpurified LS has also been reported to result in a resin with lower reactivity, resulting in increased pressing times. Therefore, LS, as the main component of adhesives, should be accompanied by a suitable chemical crosslinker in order to compensate for this lower reactivity. Another approach is the modification of the parameters for the production of new composites [101]. The use of a crosslinker could also increase water resistance [91].

Oxidized ammonium LS combined with polyethylenimine (PEI) have been studied with satisfying results as a binder for wood-based green composites. The effects of hot-pressing temperature, hot-pressing time, binder content, and MAL/PEI weight ratio on the physico-mechanical properties of the composites were investigated. The optimum operating conditions were 170 °C, 7 min, 20% *w*/*w*, 7:1, respectively. Under these conditions, the composites met the mechanical property requirements for furniture grade medium density fiberboard (MDF-FN REG). Dynamic mechanical analysis (DMA), scanning electron microscopy (SEM), FTIR, and X-ray diffraction analysis (XRD) were used to determine the influence of LS oxidation. FTIR analysis of unmodified and oxidized LS confirmed the higher reactivity of the latter. The authors also obtained evidence for the formation of amide linkages in the modified LS-PEI mixture under optimum conditions. SEM analysis confirmed the superior bonding strength of the composites obtained with modified LS. The higher crystallinity of these composites and good viscoelastic properties were also verified by XRD and DMA, respectively [84].

Composites which met the Chinese requirements for MDF have also been produced using chitosan and ammonium LS as a binder. The 6% *w*/*w* of chitosan-lignin adhesive content was used with a lignin/chitosan weight ratio of 1:2. The SEM analysis confirmed the adhesive’s high performance while the FTIR analysis indicated that hydrogen bonding between chitosan and LS occurred. Lastly, TGA proved that these linkages would decompose below 230 °C [93].

Hemmilä and co-workers also highlighted the need for need for a suitable crosslinker for LS adhesives for PBs, when LS is used without modification. Both bio-based furfuryl alcohol (FOH) and pMDI were studied as crosslinkers for ammonium LS. The addition of mimosa tannin to LS before crosslinking was also studied. A significant decrease in curing temperature and curing heat with crosslinker addition was confirmed. The best results were obtained for the LS-pMDI sample. FOH resulted in PBs with inferior mechanical properties when compared to those with pMDI, both with and without tannin. Formaldehyde emissions of pMDI crosslinked particleboards were at the level of natural wood, unlike FOH crosslinked boards which emitted beyond this level. Regarding the addition of mimosa tannin (10% to LS amount), no significant effects on the mechanical properties of PBs were observed. However, a decrease in the formaldehyde emissions of FOH crosslinked LS samples was noted. The authors concluded that, until more suitable crosslinkers are found, pMDI is a suitable option [70]. 

Polymeric isocyanate (pMDI) has been used as a crosslinker for glyoxylated LS. For methylolated lignin, the reaction mechanism with pMDI in aqueous solution has been elucidated by Pizzi. Thus, pMDI reacts preferably with the methylol groups in methylolated lignin, forming urethane bonds. Additional crosslinking occurs through the reaction of the isocyanate groups with the -NH- of the urethane bridges [76,78].

As previously mentioned, Mansouri and co-workers have replaced formaldehyde (LD50 rat > 800 mg/kg) in lignin-based wood adhesives with glyoxal, a less toxic aldehyde (LD50 rat > 2960 mg/kg), which is non-volatile with regard to the aqueous phase [78,79,80]. For this purpose, an adhesive composed of glyoxalated calcium LS and pMDI was proposed. The optimum operating conditions for PBs production were resin load on dry wood of 8%, pressing time of 28.1 s/mm board thickness at temperatures between 195–200 °C. The ratio of glyoxalated LS to pMDI was of 60 to 40. These PBs yielded good IB strength results that comfortably passed those required in international standard specifications for exterior-grade panels. Their performance was also similar to PBs manufactured with formaldehyde-based commercial adhesives [78].

LS and tannin adhesives have also been suggested. Rhazi et al. combined glyoxylated ammonium LS with natural tannin extracted using different processes to produce an adhesive for plywood panels. The optimized operating conditions for plywood production were ratio of tannin solution to glyoxylated LS of 75:25, pressing time of 4 min, temperature of 150 °C, and pressure of 12 bar. The produced panels showed similar mechanical properties to those made with the commercial UF and PF resins. However, formaldehyde emission levels were also significantly lower than when commercial PF resins were used. These panels also followed the requirements of EN 314-2:1993, and therefore presented good bonding quality [94]. 

## 6. Conclusions and Future Challenges

Although lignin is the most abundant renewable source of phenolic compounds of natural origin, it is usually treated as an industrial sub-product and burned as fuel. Lignosulphonates, obtained from the sulphite pulping process, alone account for 90% of the total market of commercial lignin with a total annual global production of approximately 1.8 million tons.

As the chemical structure and composition of lignin varies with the source and isolation method, it is important to carry out a detailed characterization before it can be included in the production of value-added products. Currently, there are no uniform or standardized methods for the characterization of lignin. Nevertheless, efforts are being made to develop a series of ISO methods for the characterization of the following lignin features: general composition; functional groups; size and morphology; thermal properties; structural features; safe handling; and processability. Thus, there is little information on fundamental analysis and characterization of technical lignins in the literature, especially when it comes to lignosulphonates. Additionally, the methods that are currently applied need to be adapted before they can be used on lignosulphonates.

One of the possible applications of lignins and lignosulphonates in the production of wood adhesives. The most successful industrial attempts to use lignin for this purpose are based on using this polymer as a partial substitute in phenol-formaldehyde resins or urea-formaldehyde resins. Although lignin and phenol present a similar structure, lignin presents a lower number of reactive sites for polymerization reactions.

In order to address this issue, several methods to improve the reactivity of lignin molecules have been studied with the most promising being phenolation and hydroxymethylation. While these methods influence the reactivity of lignin in resin synthesis, they rely on organic reagents. Thus, there is still a need for more eco-friendly methods like oxidation and ionic liquid treatment.

So far LPF resins with 35% *w*/*w* of replaced phenol by methylolated softwood ammonium LS which complied with the standards for plywood application and displayed similar characteristics to commercial PF resins have been developed. In other studies, high-density fiberboard panels were produced with low content UF, 3%, and ammonium LS content of 8%, based on the dry fibers which complied with the requirements for use in loadbearing applications in humid conditions.

However, these resins include formaldehyde which is a known carcinogenic. Thus, concerns about formaldehyde emissions and the sustainability of the used raw materials and final products are reducing the popularity of formaldehyde-based synthetic resins for wood-based panels. Formaldehyde emissions from panels, especially in indoor applications, are raising great concerns.

Therefore, bio-based adhesives have been increasingly studied. Systems which combine LS with polyethylenimine, furfuryl alcohol, polymeric isocyanate, chitosan, tannins, glyoxal, and wheat flour have been proposed.

Nevertheless, lignin adhesives are associated with long pressing times due to their low reactivity, which increases production costs in panel manufacturing. Studies have also reported the low water resistance of the obtained wood panels. Thus, to address these problems and increase the industrial success of lignin and LS adhesives, the interaction between these compounds needs to be further elucidated and more suitable crosslinkers need to be studied.

To conclude, although several studies have been conducted on lignosulphonates, important advances are still needed to warrant success for lignin-based adhesives, namely in their characterization techniques, methods for improving reactivity, and selection of suitable crosslinkers.

## Figures and Tables

**Figure 1 polymers-13-04196-f001:**
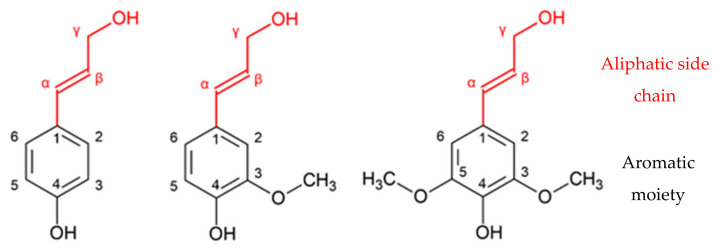
Monomeric lignin precursors: *p*-coumaryl alcohol, coniferyl alcohol, and sinapyl alcohol (adapted from [6]).

**Figure 2 polymers-13-04196-f002:**
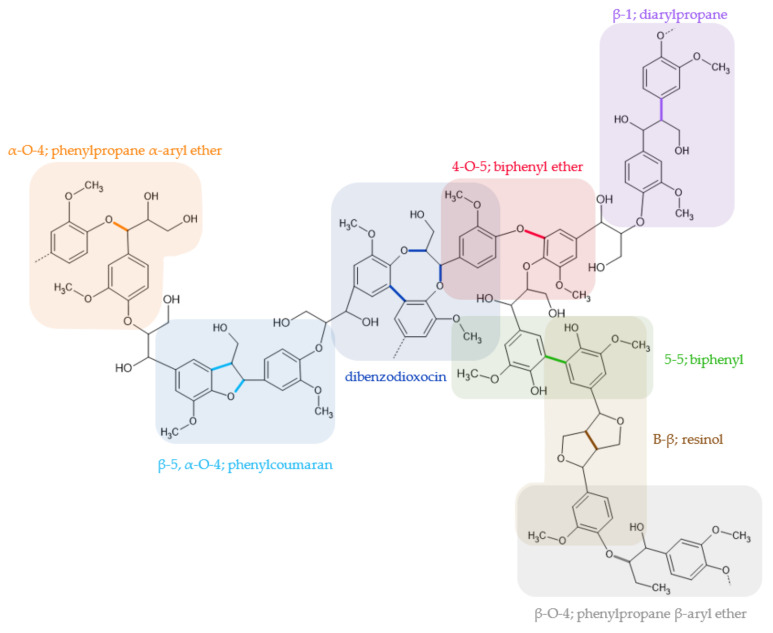
Model of lignin’s structure with the important linkages and units (adapted from [7]).

**Figure 3 polymers-13-04196-f003:**
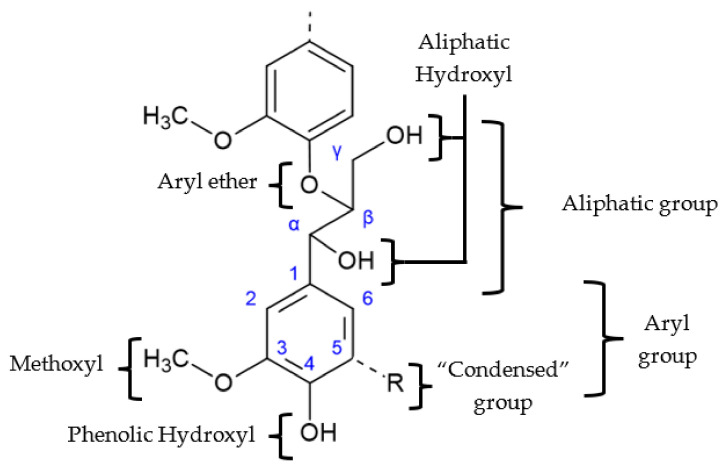
Main functional groups in lignin’s structure (adapted from [14]).

**Figure 4 polymers-13-04196-f004:**
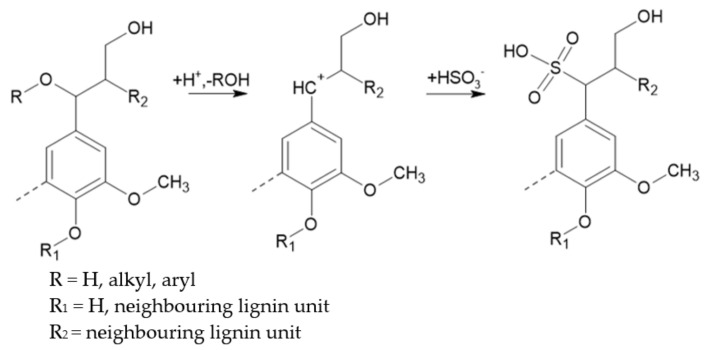
Main reactions for lignosulphonate formation during acid sulphite pulping (adapted from [16]).

**Figure 5 polymers-13-04196-f005:**
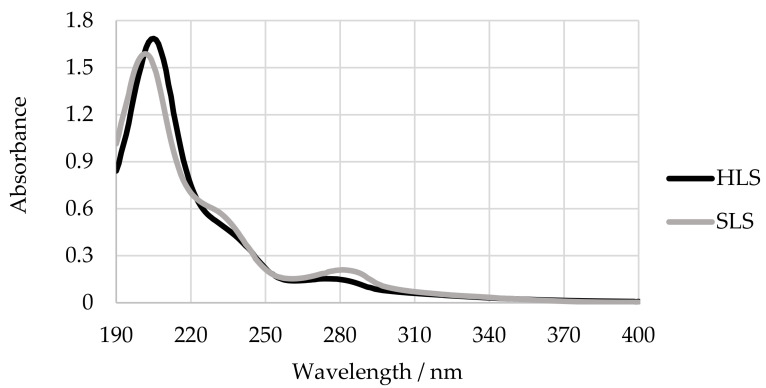
UV spectra of hardwood magnesium-based thick spent sulphite liquor (HLS) and softwood sodium lignosulfonates (SLS) in water.

**Figure 6 polymers-13-04196-f006:**
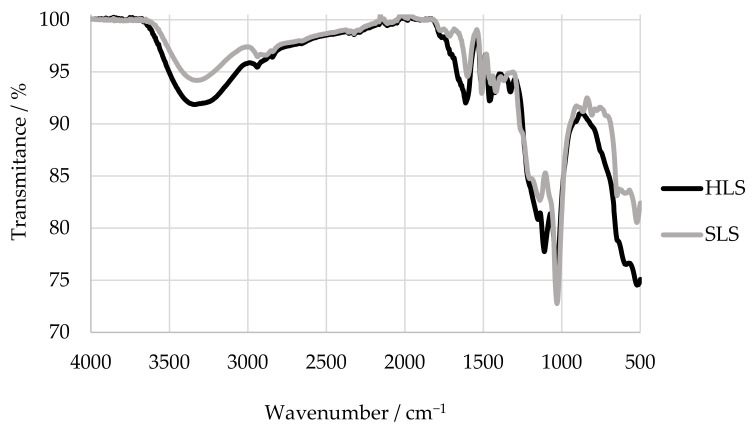
FTIR spectra of HLS and SLS.

**Figure 7 polymers-13-04196-f007:**
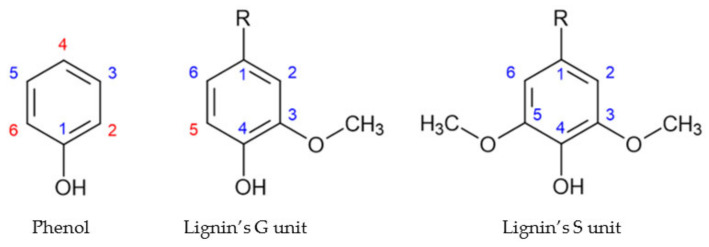
Reactive sites of Phenol and lignin’s phenolic units (adapted from [23]).

**Figure 8 polymers-13-04196-f008:**
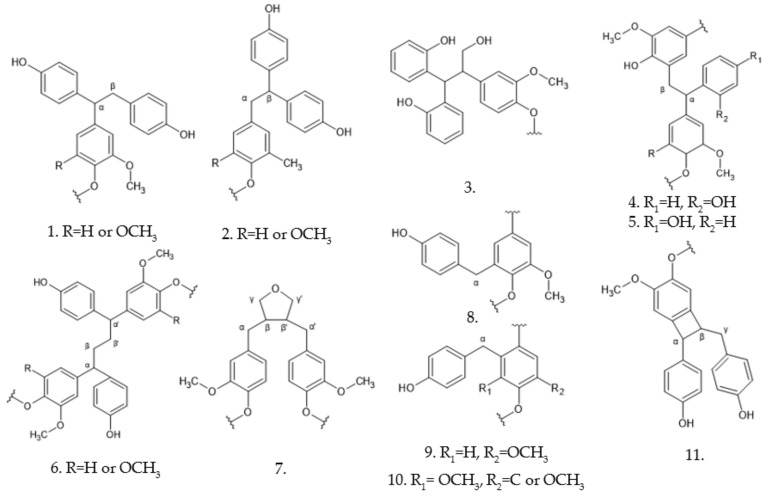
Possible structures present in phenolated lignin (adapted from [71]).

**Figure 9 polymers-13-04196-f009:**
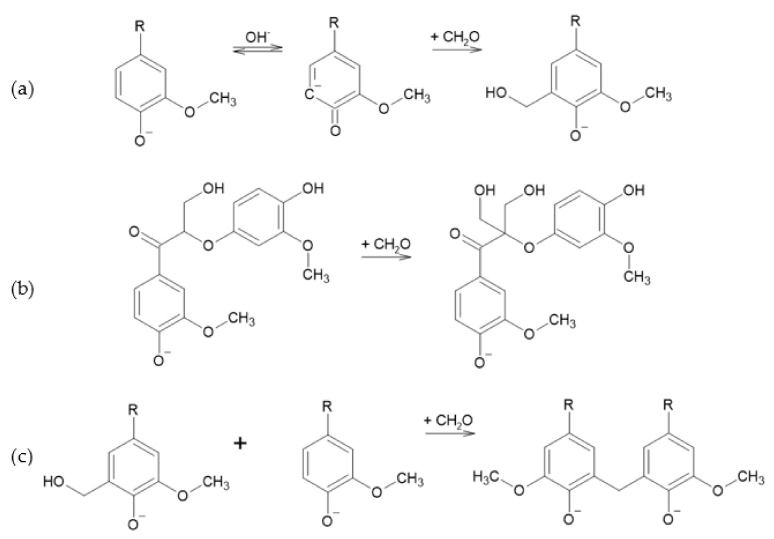
The Lederer–Manasse (**a**), Tollens (**b**), and further condensation reactions (**c**) (adapted from [75]).

**Figure 10 polymers-13-04196-f010:**
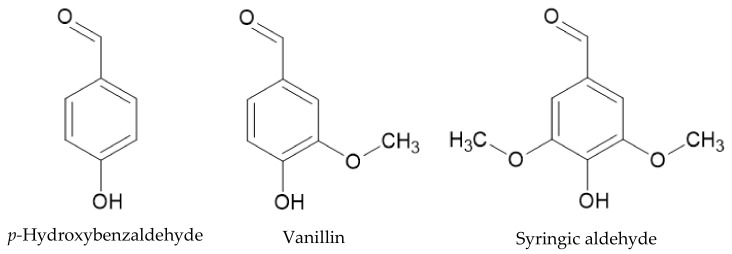
Aromatic aldehydes derived from lignin’s oxidation (adapted from [83]).

**Figure 11 polymers-13-04196-f011:**
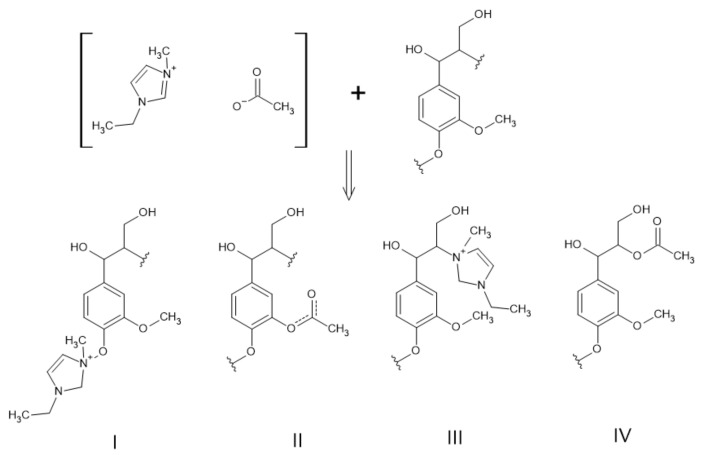
Possible reactions between lignin and [Emim][OAc] (adapted from [89]).

**Figure 12 polymers-13-04196-f012:**
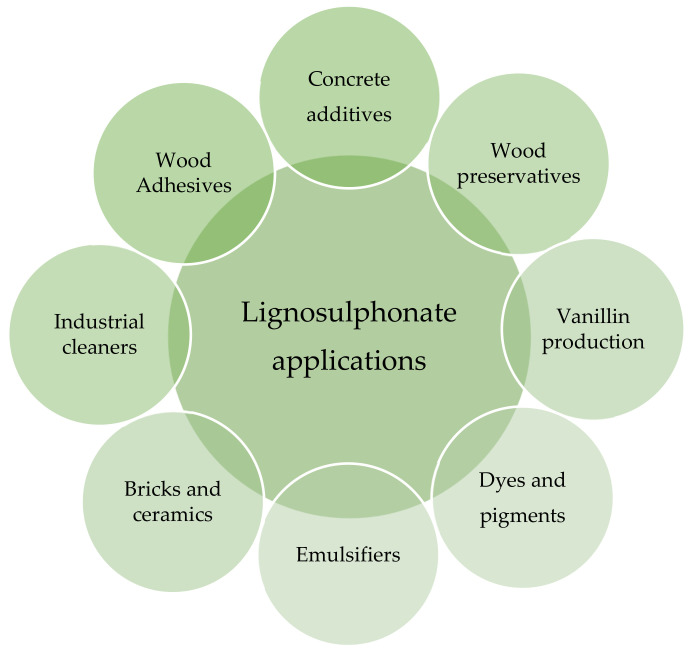
LS applications.

**Table 1 polymers-13-04196-t001:** Percentages of the monolignols in lignin in different plants [9].

Linkage Type	*p*-Coumaryl Alcohol (%)	Coniferyl Alcohol (%)	Sinapyl Alcohol (%)
Coniferous; softwoods	<5 ^a^	>95	0 ^b^
Eudicotyledonous; hardwoods	0–8	25–50	45–75
Monocotyledonous; grasses	5–35	35–80	20–55

^a^ Higher amount in compression wood. ^b^ Some exceptions exist.

**Table 2 polymers-13-04196-t002:** Types and frequencies of linkages in softwood and hardwood lignins [11].

Linkage Type	Softwood (Spruce) (%)	Hardwood (Birch) (%)
*β-*O-4-Aryl ether	46	60
α-O-4-Aryl ether	6–8	6–8
4-O-5-Diaryl ether	3.5–4	6.5
*β*-5-Phenylcoumaran	9–12	6
5-5-Biphenyl	9.5–11	4.5
*β*-1-(1,2-Diarylpropane)	7	7
*β-β*-(Resinol)	2	3
Others	13	5

**Table 3 polymers-13-04196-t003:** Different modifications of the sulphite method [13,18].

	Acid (bi)Sulphite	Bisulphite	Neutral Sulphite	Alkaline Sulphite
pH range	1–2	3–5	6–9	9–13
Base alternatives	Ca^2+^, Mg^2+^, Na^+^, NH_4_^+^	Mg^2+^, Na^+^, NH_4_^+^	Na^+^, NH_4_^+^	Na^+^
Active reagents	HSO_3_^−^, H^+^	HSO_3_^−^, H^+^	HSO_3_^−^, SO_3_^2−^	SO_3_^2−^, HO^−^
Max. temp. (°C)	125–145	150–170	160–180	160–180
Time at max. temp. (h)	3–7	1–3	0.25–3	3–5
Softwood pulp yield (%)	45–55	50–65	75–90 ^a^	45–60

^a^ Hardwood.

**Table 4 polymers-13-04196-t004:** Main chemical properties of technical lignins [20].

Parameter	Lignosulphonates	Kraft Lignin	Soda Lignin	Organosolv Lignin
Ash content (%)	4.0–8.0	0.5–3.0	0.7–2.3	1.7
Sulphur content (%)	3.5–8.0	1.0–3.0	0	0
Molecular weight, Mw	1000–50,000 (up to 150,000)	1500–5000 (up to 25,000)	1000–3000 (up to 15,000)	500–5000
Polydispersity	4.2–7.0	2.5–3.5	2.5–3.5	1.5

**Table 5 polymers-13-04196-t005:** GPC systems for lignosulphonates and spent sulphite liquors (adapted from [39]).

Sample	Column Type	Eluent	Standards	Detectors	Reference
**Organic Solvent**					
LS and LS–QAM complex	3 SDVB (styrene–divinylbenzene) columns	THF + QAM (quaternary amine methyltrioctylammonium chloride)	PS and biphenyl	UV	[47]
SSLs fractions	PSS GRAM 30, 2 columns and a guard column	LiBr (0.05 M) in DMSO/ water (90:10)	Pullulan	UV RI Viscosimetric	[48]
SSLs fractions	2 Polyacrylate methacrylate Columns	DMSO:H_2_O (9:1) and 0.05 M LiBr	Pullulan (high Mw) glucose/cellobiose (low Mw)	RI Viscosimetric	[42]
	2 PFG-PRO (silica)	DMAc and 0.11 M LiCl	Polyethylene glycol Polyethylene oxide		
Purified LS	3 Agilent PolarGel M colums 1 guard column	DMSO/LiBr (0.5% *w*/*v*)	PSS	UV RI	[41]
**Aqueous System**					
SSL and purified LS	2 PL aquagel And pre-column	NaNO_3_ 0.1 M	PS	RI	[49]
LS and fractioned LS	Jordi Glucose-DVB And pre-column	Water/DMSO/ Na_2_HPO_4_-4H_2_O/SDS	-	DAWN-F MALLS RI	[46]
LS	Ultrahydrogel or Ultrastyragel	Na_2_NO_3_ or NaCl solutions	Pulluan	RI	[50]

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
