# Peer review of "Lignosulphonates as an Alternative to Non-Renewable Binders in Wood-Based Materials"

_polymers, 2021, doi:10.3390/polym13234196_

Round 1
Reviewer 1 Report
Review report for manuscript "polymers-1487510" Lignosulphonates as an alternative to non-renewable binders in wood-based materials. This manuscript was excellently written, I have only a few minor suggestions for improving it:
In the Abstract, please discuss more results of your research, implications for practice, and perspectives on using LS in wood-based materials. The first half of the Abstract is almost the same as in the first two paragraphs in the Introduction part (without sources).
Line 40: The information about "lignin is the most abundant..." is already in the abstract, then again in line 27.
Line 107: no need to define again abbr. LS was defined in line 14.
Please check the English throughout the manuscript.
Reviewer 2 Report
The presented manuscript is a comprehensive review article of the potential of using lignosulfonates as lignin-based, formaldehyde-free adhesives for bonding wood-based panels. The manuscript is well-structured, informative and provides relevant information and references on the topic. Some of the most recent findings and developments in the field of lignosulfonate-based wood adhesives have been properly presented and discussed. Congratulations to the authors!